# A Discrete Actor and Critic for Reinforcement Learning on Continuous Tasks

## Abstract

Solving continuous reinforcement learning (RL) tasks typically requires models with continuous action spaces, as discrete models face challenges such as the curse of dimensionality. Inspired by discrete controlling signals in control systems, such as pulse-width modulation, we investigated RL models with discrete action spaces with performance comparable to continuous models on continuous tasks. In this paper, we propose an RL model with a discrete action space, designed a discrete actor that outputs action distributions and twin discrete critics for value distribution estimation. We also developed both the training method and exploration strategy for this model. The model successfully solved BipedalWalkerHardcore-v3, a continuous robot control task in a complex environment, achieved a higher score than the state-of-the-art baselines and comparable results across various other control tasks.

## 1 Introduction

Reinforcement Learning (RL) typically applies discrete action space for discrete tasks and continuous action space for continuous tasks. For discrete tasks, such as Atari games, Q-learning and its variances exhaustively evaluate only a small number of actions. For continuous tasks, such as motion control tasks, evaluating all possible action is not possible, hence RL models with discrete action space can suffer from the curse of dimensionality (Kober et al., 2014; Lillicrap et al., 2016). To avoid this problem, models for continuous tasks directly output actions with continuous values (Lillicrap et al., 2016; Schulman et al., 2017; Fujimoto et al., 2018; Haarnoja et al., 2018; Kuznetsov et al., 2020).

However, in the context of the control system, using discrete signals for control is widely adopted. For example, pulse-width modulation (PWM) is used for motor control and light control. PWM represents signals with zeros and ones (sometimes also with -1), which is purely discrete. However, by varying the ratio of the time discrete values are presented, PWM can approximate continuous signals (Figure 1) (Yu et al., 1997) and control motors precisely (Hughes and Drury, 2013). For light-emitting diode (LED), because of its nonlinear relations between its luminous intensity and input voltage, control with continuous voltage is even harder than with PWM controlled voltage (Esteki et al., 2023). PWM linearises the relations by switching either on or off the light and controls intensities by the time ratio, simplifying the control complexity. Hence, the discrete signals have been proved useful for control tasks.

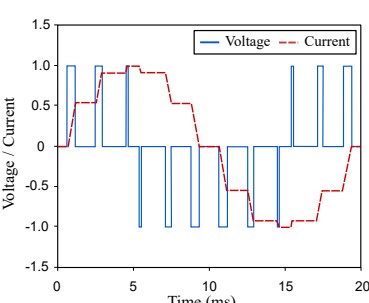

Figure 1: Discrete voltages applied on an indicator results in continues current.

Based on this fact, a reinforcement learning model with a discrete action space can perform similarly on continuous tasks as continuous models. With idea, we proposed a model with discrete action space for continuous tasks.

We designed a discrete actor module, modified a Categorical DQN (C51) (Bellemare et al., 2017), proposed twin discrete critic networks inspired by TD3 (Fujimoto et al., 2018), and composed them together. The actor outputs the probability of action atoms which are from even discretization of actions. C51 is an example of Distributional Reinforcement Learning (Distributional RL) which can be used in modeling the distribution of reward returns of partially observable problems. It adopts

discrete value distribution and updates the probability of value categories instead of fitting a curve, hence it can withstand strong reward impacts.

The model can solve various continuous tasks with performance near state-of-the-art (SOTA) models. For BipedalWalkerHardcore-v3, with discrete value distribution, our model with discrete action space achieve higher scores than SOTA models. Our model achieved an average score of 324.8 on 10,000 trials, the highest currently (May. 2024) on the Leaderboard (OpenAI).

## 2 RELATED WORK

The following Deep RL algorithms have been applied in continuous tasks. DDPG (Lillicrap et al., 2015) effectively applies Q-learning principles to continuous action spaces for Deep RL. TD3 (Fujimoto et al., 2018) an extension of DDPG, utilizes twin critic networks to address value overestimation. SAC (Haarnoja et al., 2018) employs stochastic continuous actor and entropy regularization. TQC (Kuznetsov et al., 2020), building upon QR-DQN (Dabney et al., 2018b), integrates an actor and employs a truncated mixture mechanism to mitigate value overestimation. DDPG, TD3, SAC, TQC, and PPO have demonstrated commendable performance in MuJoCo tasks (Towers et al., 2023). Stable Baselines 3 (Raffin et al., 2021) and CleanRL (Huang et al., 2022) offer reliable implementations of these fundamental Deep RL algorithms, and we used some of them for comparison in our experiments.

Distributional RL models focus on modeling the distribution of cumulative rewards rather than only an expected scalar. C51 (Bellemare et al., 2017) employs a discrete value distribution for building its critic network. QR-DQN (Dabney et al., 2018b) and IQN (Dabney et al., 2018a) utilize quantile regression to detail the distribution of stochastic reward returns. These methodologies, despite their varied mathematical constructs for their critic networks, predominantly affirm theoretical convergence under Wasserstein Metric (Vaserstein, 1969) and have notable performance on Atari tasks.

Some previous works resemble a few aspects of our work but differently. D4PG (Barth-Maron et al., 2018) proposed a C51 with an actor module. Our work, however, diverges by investigating discretized action spaces, moving away from the conventional assumption that action variables conform to a normal distribution. SAC-Discrete (Christodoulou, 2019) broadens the scope of SAC to discrete action spaces, thus enhancing the model's capacity to use action entropy for exploration.

There were explorations connecting continuous action Neunert et al. (2020) explores a feasible approach for the unified control of discrete and continuous action variables based on the MPO algorithm. Metz et al. (2019) decomposes multi-dimensional action variables into a sequence of decision-making processes for discrete variables, though this method risks oversimplifying complex tasks and can lead to increased computational demands. Tang and Agrawal (2020) advocates for the discretization of continuous action spaces, which can enhance the performance of on-policy algorithms such as PPO, albeit at the potential expense of capturing continuous dynamics accurately. Luo et al. (2023) emphasizes the benefits of discretizing action spaces in offline reinforcement learning and examines potential solutions, but highlighted data and computational load challenges, impacting real-world performance. Farebrother et al. (2024) argues that the cross-entropy loss function, compared to the mean squared error loss function, is more effective for training Critic networks in reinforcement learning, especially for models with large parameter counts like Transformers. However, it presents difficulties in transitioning to environments that require continuous representations.

## 3 PRELIMINARIES

This section covers relevant prior work and essential concept basis in our model. To maintain consistency, the mathematical symbols in this paper align closely with those used in (Bellemare et al., 2017), which has some notations with a style different from typical machine learning. For convenience, we provide Table 3 for a reference of symbols.

### 3.1 DISCRETE VALUE DISTRIBUTION

For a stochastic transition process $(\boldsymbol{x}, \boldsymbol{a}) \to (\hat{\boldsymbol{x}}', \hat{\boldsymbol{a}}')$ in an environment, $\boldsymbol{x}$ represents the observed current state of the environment, and $\boldsymbol{a}$ specifies the action taken in response to $\boldsymbol{x}$. The resulting

state distribution is denoted $\hat{\boldsymbol{x}}'$. A stochastic policy output an action distribution $\hat{\boldsymbol{a}}'$, and the actual action $\boldsymbol{a}'$ taken in the task will be sampled from $\hat{\boldsymbol{a}}'$. The value $Z$ associated with the process $(\boldsymbol{x}, \boldsymbol{a}) \rightarrow (\hat{\boldsymbol{x}}', \hat{\boldsymbol{a}}')$ is formularized using a recursive equation: $Z(\boldsymbol{x}, \boldsymbol{a}) \overset{D}{=} R(\boldsymbol{x}, \boldsymbol{a}) + \gamma Z(\hat{\boldsymbol{x}}', \hat{\boldsymbol{a}}')$, where $R(\boldsymbol{x}, \boldsymbol{a})$ represents the stochastic reward function of the environment and $\gamma$ denotes the discount rate.

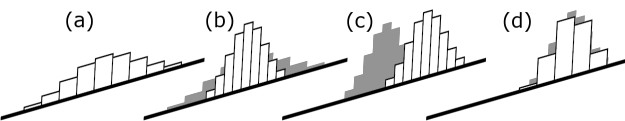

Figure 2: Operations to update $Z$. (a) The current value distribution of $Z$. (b) Discount factor $\gamma$ changes the shape in the dimension of atoms. (c) The current reward $R$ shifts the distribution in the dimension of atoms. (d) The resulting distribution $R + \gamma Z$ is mapped back to the original atoms by $\Phi$. Adapted from (Bellemare et al., 2017)

In the Categorical DQN, the value $Z$ is conceptualized as a random variable with a discrete value distribution. To update the distribution, Bellemare et al. (2017) proposed an updating rule (Figure 2). The number of discrete atoms $N \in \mathbb{N}$ denotes the granularity of discretization required for the value domain, and the bounds $V_{MIN}, V_{MAX} \in \mathbb{R}$ specify the lower and upper limits of the values, respectively. The set of discrete atoms is constructed as $\{z_i = V_{MIN} + (i-1)\triangle z | i = 1, 2, \cdots N\}$, with the interval $\triangle z$ calculated by $\frac{V_{MAX} - V_{MIN}}{N-1}$. The probability of each discrete atom's occurrence is determined using a neural network $\Theta : \mathcal{X} \times \mathcal{A} \rightarrow \mathbb{R}^N$, that is, $Z(\boldsymbol{x}, \boldsymbol{a} | \Theta) = z_i$ $\quad w.p. \quad p_i(\boldsymbol{x}, \boldsymbol{a}) = \frac{e^{(\Theta(\boldsymbol{x}, \boldsymbol{a}))_i}}{\sum_j^N e^{(\Theta(\boldsymbol{x}, \boldsymbol{a}))_j}}$. For a tuple of a stochastic transition $\boldsymbol{t} = (\boldsymbol{x}, \boldsymbol{a}, r, \boldsymbol{x}')$, a Bellman update is applied to each discrete atom $z_j$, designated as $\hat{\mathcal{T}} z_j := r + \gamma z_j$. The probability associated with $\hat{\mathcal{T}} z_j$, denoted $p_j(\boldsymbol{x}', \pi(\boldsymbol{x}'))$, is then redistributed amongst adjacent discrete atoms. The $i^{\text{th}}$ element of the resultant projected discrete probability distribution $\Phi \hat{\mathcal{T}} Z(\boldsymbol{x}, \boldsymbol{a} | \Theta)$ is: $P(\Phi \hat{\mathcal{T}} Z(\boldsymbol{x}, \boldsymbol{a} | \Theta) = z_i) = \sum_{j=1}^N \left[ 1 - \frac{|[\hat{\mathcal{T}} z_j]_{V_{MIN}}^{V_{MAX}} - z_i|}{\triangle z} \right]_0^1 p_j(\boldsymbol{x}', \pi(\boldsymbol{x}'))$ The notation $[\cdot]_a^b$ signifies that the value is constrained within the interval $[a, b]$. For an elaborate exposition and validation of these concepts, please consult the original paper (Bellemare et al., 2017).

### 3.2 CLIPPED DOUBLE Q-LEARNING FOR ACTOR-CRITIC

TD3 (Fujimoto et al., 2018) adopts Double Q-learning to reduce the overestimation bias of Q-value. TD3 is composed of three networks: $Q_{\psi_1}(\boldsymbol{x}, \boldsymbol{a})$, $Q_{\psi_2}(\boldsymbol{x}, \boldsymbol{a})$, and $\pi_\phi(\boldsymbol{x})$. The twin Q networks, $Q_{\psi_1}(\boldsymbol{x}, \boldsymbol{a})$ and $Q_{\psi_2}(\boldsymbol{x}, \boldsymbol{a})$, though initially having distinct parameters, are trained concurrently with the same learning signals. $\pi_\phi(\boldsymbol{x})$ serves as the actor network. The TD3 update procedure for a data batch involves three steps:

First, utilize the twin networks for a more precise estimate of the reward return, denoted as $y$. $y \leftarrow r + \gamma \min_{i=1,2} Q_{\psi_i'}(\boldsymbol{x}', \pi_{\phi'}(\boldsymbol{x}') + \boldsymbol{\epsilon})$, where $\psi_i'$ is the $i^{\text{th}}$ target critic network's delayed updated parameters, $\phi'$ is the Target actor network's delayed updated parameters, and $\boldsymbol{\epsilon}$ is minor noise.

Then apply the mean squared error loss function to align the twin Q networks with the corrected Q-value $y$, thus the $i^{\text{th}}$ critic network's parameters $\psi_i$ is updated following $\psi_i \leftarrow \psi_i - \frac{\alpha}{B} \sum_{\boldsymbol{t} \sim \mathcal{D}} \nabla_{\psi_i} (y - Q_{\psi_i}(\boldsymbol{x}, \boldsymbol{a}))^2$, where $\alpha$ is the learning rate, $B$ is the batch size, and $\boldsymbol{t} \sim \mathcal{D}$ signifies a batch of transition samples from the Replay Buffer, $\boldsymbol{t} = (\boldsymbol{x}, \boldsymbol{a}, r, \boldsymbol{x}')$.

Finaly, the actor network, $\pi_\phi(\boldsymbol{x})$, learns based on $Q_{\psi_1}(\boldsymbol{x}, \boldsymbol{a})$ independently of $Q_{\psi_2}(\boldsymbol{x}, \boldsymbol{a})$, where $\phi$ denotes the actor network's parameters: $\phi \leftarrow \phi + \frac{\alpha}{B} \sum_{\boldsymbol{t} \sim \mathcal{D}} \nabla_\phi \pi_\phi(\boldsymbol{x}) \nabla_{\boldsymbol{a}} Q_{\psi_1}(\boldsymbol{x}, \pi_\phi(\boldsymbol{x}))|_{\boldsymbol{a} = \pi_\phi(\boldsymbol{x})}$.

# 4 MODELS & METHODS

Our model (Figure 3) extends previous work by addressing its limitations to improve the algorithm's performance and robustness by multidimensional discrete action space, clipped double Q-learning for discrete value distribution, corresponding learning rule for critic and actor, and balancing exploration and exploitation.

## 4.1 MULTIDIMENSIONAL DISCRETE ACTOR

Our model is based on discrete action space across multiple dimensions. A one-dimensional continuous action space is discretized into $m$ discrete action atoms $\{a_1, a_2, \cdots, a_m\}, m \in \mathbb{N}$, where $\mathbb{N}$ denotes the set of natural numbers. Then the discretization is applied to each dimension of a $n$-dimensional continuous action space, so an action in this new discrete space $\mathcal{A}$ can be noted as a matrix $A \overset{D}{=} [\boldsymbol{a_1}, \boldsymbol{a_2}, \cdots, \boldsymbol{a_n}]^\mathsf{T}$, where each row is one-hot coding of the corresponding action dimension. This shape is convenient to match the action probability distribution $\hat{A}$, which will be used as the output of policy network, and the sum of each row of $\hat{A}$ is 1. When

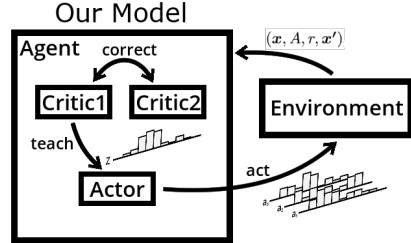

Figure 3: Algorithmic Framework Overview

we sample $A$ from $\hat{A}$, each row in $A$ is sampled from the probability of the corresponding row in $\hat{A}$.

In this action space, there exist $m^n$ discrete potential actions. Given such an extensive search space, employing exhaustive search methods such as those used by traditional DQN algorithms to find the maximal Q-value is not feasible. In this study, we propose modeling the agent's stochastic behavior within the action space $\mathcal{A}$ by utilizing an action probability matrix, therefore we set the actor as $\pi : \mathcal{X} \to \mathbb{R}^{n \times m}$, that is,

$$
\pi(\boldsymbol{x}) \overset{D}{=} \begin{bmatrix} p_{11}(\boldsymbol{x}) & p_{12}(\boldsymbol{x}) & \cdots & p_{1m}(\boldsymbol{x}) \\ p_{21}(\boldsymbol{x}) & p_{22}(\boldsymbol{x}) & \cdots & p_{2m}(\boldsymbol{x}) \\ \vdots & \vdots & \ddots & \vdots \\ p_{n1}(\boldsymbol{x}) & p_{n2}(\boldsymbol{x}) & \cdots & p_{nm}(\boldsymbol{x}) \end{bmatrix},
\tag{1}
$$

where $p_{ij}(\boldsymbol{x}) \geq 0$, $\sum_{j=1}^m p_{ij}(\boldsymbol{x}) = 1$ for $i = 1, 2, \cdots, n$, and $\boldsymbol{x}$ is an observed state. This $\pi$ characterizes a stochastic multi-dimensional discrete actor. A later section will detail how to use a neural network to approximate $\pi$. Please note that, in the original continuous action space, the action dimensions are independent, so in $A$, the elements between rows are also independent.

## 4.2 CLIPPED DOUBLE Q-LEARNING FOR DISCRETE VALUE DISTRIBUTION

We discussed TD3's approach to mitigating overestimation by Double Q-learning in the earlier sections. Here we modify it for the discrete value distribution.

Double Q-learning uses two critic networks, $\Theta_{\psi_1}(\boldsymbol{x}, \hat{A})$ and $\Theta_{\psi_2}(\boldsymbol{x}, \hat{A})$, and one actor network $\pi_\phi(\boldsymbol{x})$. It also has target networks $\Theta_{\psi_1'}(\boldsymbol{x}, \hat{A})$, $\Theta_{\psi_2'}(\boldsymbol{x}, \hat{A})$, and $\pi_{\phi'}(\boldsymbol{x})$ correspond to the main networks for stability in training. The subscripts above, $\psi_1, \psi_2, \phi, \psi_1', \psi_2'$, and $\phi'$, denote the parameters of corresponding networks. Given a transition tuple $\boldsymbol{t} = (\boldsymbol{x}, A, r, \boldsymbol{x}')$, we consider how to effectively utilize these target networks to yield an updated estimation of the value distribution, $\Phi \hat{\mathcal{T}} \tilde{Z}(\boldsymbol{x}, \hat{A} | \Theta_{\psi_1'}, \Theta_{\psi_2'})$. With $\Theta_{\psi_1'}$ and $\Theta_{\psi_2'}$, for $\hat{A}' = \pi_{\phi'}(\boldsymbol{x}')$, we derive $\Phi Z(\boldsymbol{x}', \hat{A}' | \Theta_{\psi_i'})$ as:

$$
P(Z(\boldsymbol{x}', \hat{A}' | \Theta_{\psi_i'}) = z_k) \overset{D}{=} \frac{e^{(\Theta_{\psi_i'}(\boldsymbol{x}, \hat{A}'))_k}}{\sum_j^N e^{(\Theta_{\psi_i'}(\boldsymbol{x}, \hat{A}'))_j}}
\tag{2}
$$

A set of procedures (Figure 4) is proposed to leverage the twin critic networks with a discrete value distribution. (a) Firstly, the two critic networks estimate discrete value distributions according to

$x'$, respectively. (b) Secondly, the distributions are accumulated respectively. (c) Then for each category across the cumulative distributions, the one with a higher probability is selected to form a new cumulative distribution, by which the value is less susceptible to overestimation. (d) Finally, each category of the new cumulative distribution, except the first one, is subtracted by the former one, mapping it back to discrete value distribution.

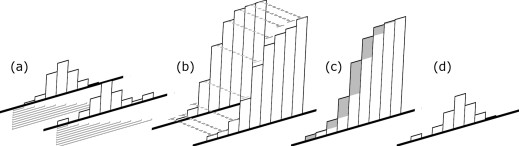

Figure 4: Clipped Double Discrete Value Distribution

In procedure (b) cumulative the distribution in (a). For a discrete value distribution $Z$, $P(Z \in \{z_1, z_2, \cdots, z_k\}) = \sum_{j=1}^{k} P(Z = z_j)$. For the $k^{\text{th}}$ value atom, the third and fourth procedures can be presented as:

$$P(\tilde{Z}(x', \hat{A}'|\Theta_{\psi'_1}, \Theta_{\psi'_2}) = z_k) = \begin{cases} c_k & \text{if} \quad k = 1 \\ c_k - c_{k-1} & \text{if} \quad k > 1 \end{cases} \tag{3}$$

where $c_k = \max_{i=1,2} P(Z(\boldsymbol{x'}, \hat{A}'|\Theta_{\psi'_i}) \in \{z_1, z_2, \cdots, z_k\})$. For the transition sample $\boldsymbol{t} = (\boldsymbol{x}, A, r, \boldsymbol{x'})$ and the $i^{\text{th}}$ value atom, the Bellman Operation is as follows:

$$P(\Phi\hat{\mathcal{T}}\tilde{Z}(\boldsymbol{x}, A|\Theta_{\psi'_1}, \Theta_{\psi'_2}) = z_i) = \sum_{j=1}^{N} \left[ 1 - \frac{|[\hat{\mathcal{T}}z_j]_{V_{MIN}}^{V_{MAX}} - z_i|}{\triangle z} \right]_0^1 P(\tilde{Z}(\boldsymbol{x'}, \hat{A}'|\Theta_{\psi'_1}, \Theta_{\psi'_2}) = z_j) \tag{4}$$

where $A$ notes the actual action taken rather than a probability distribution, and $\Phi\hat{\mathcal{T}}\tilde{Z}(\boldsymbol{x}, A|\Theta_{\psi'_1}, \Theta_{\psi'_2})$ is the corrected discrete value distribution used in training $\Theta_{\psi_1}$ and $\Theta_{\psi_2}$.

Section 4.1 introduced the multidimensional discrete actor. The actor outputs action distribution by a probability matrix. This matrix serves as a direct input for the action component of the critic network, enabling it to model the impact of stochastic actions characterized by multidimensional discrete probability distributions. With the above discussion, the critic network's update rule for a data batch with size $B$ and $i = 1, 2$ is defined as:

$$Z_1 \stackrel{D}{=} \Phi\hat{\mathcal{T}}\tilde{Z}(\boldsymbol{x}, A|\Theta_{\psi'_1}, \Theta_{\psi'_2}), \quad Z_2 \stackrel{D}{=} Z(\boldsymbol{x}, \hat{A}|\Theta_{\psi_i})$$
$$\boldsymbol{\psi_i} \leftarrow \boldsymbol{\psi_i} - \frac{\alpha}{B} \sum_{\boldsymbol{t} \sim \mathcal{D}} \nabla_{\boldsymbol{\psi_i}} D_{KL}(Z_1||Z_2) \tag{5}$$

where $D_{KL}$ represents KL divergence, furthermore, $\nabla_{\boldsymbol{\psi_i}} D_{KL}(Z_1||Z_2) = -\sum_{j=1}^{N} P(Z_1 = z_j) \nabla_{\boldsymbol{\psi_i}} \log P(Z_2 = z_j)$. In this equation, we eliminate terms that are independent of $\boldsymbol{\psi_i}$, thus obtaining a form consistent with cross-entropy loss. $Z_1$ denotes the new estimation of the value distribution procured from the twin critic networks, and $Z_2$ is the critic network's resultant output. In this way, every critic's output is refined to align with the corrected value $Z_1$, reducing the overestimation bias.

Training a critic network to fit a categorical distribution with a cross-entropy loss is more stable than directly fitting a curve with a mean squared error loss. This phenomenon is probably because the probability in categorical distribution is confined between 0 and 1, while the reward itself can be one hundred times larger in magnitude. In the case of Bipedalwalker tasks, falling down causes a large punishment (-100), which affects the learning of the critic during walking (typically from -0.3 to 0.3). During fitting this type of multimodal distribution, each peak causes a similar impact to learning of other peaks like outliers.

### 4.3 POLICY LEARNING

We assume that the distribution of actions in a task is not a unimodal distribution, especially for tasks like BipedalWalkerHardcore-v3 which has changing terrains. Therefore, we discretize the action space as described in Section 4.1. To train the actor robustly, the actor is trained with a similar loss function for training the critic networks as introduced in Section 4.2, thereby contributing to the overall stability of the model.

Like other RL models with actor-critic architecture, the actor is updated to maximize the Q-value predicted by a critic network. Differently, in our model, the output of the critic networks is probability, so the cumulative distribution can be used as the objective. More specifically, for the $k^{\text{th}}$ value atom,

$$
\begin{aligned}
P(Z(\boldsymbol{x}, \pi_\phi(\boldsymbol{x})|\Theta_{\boldsymbol{\psi_1}}) \in \{z_1, z_2, \cdots z_k\}) &\to 0 \\
P(Z(\boldsymbol{x}, \pi_\phi(\boldsymbol{x})|\Theta_{\boldsymbol{\psi_1}}) \in \{z_{k+1}, z_{k+2}, \cdots z_N\}) &\to 1.
\end{aligned}
\tag{6}
$$

The notation "$\to$" here denotes a trend or movement toward a value. The goal is for the policy to minimize the probability of $Z$ occurring at lower-value atoms while maximizing it at higher-value atoms. With the binary cross-entropy loss applied, the Policy Learning rules are established thus:

$$
\phi \leftarrow \phi + \frac{\alpha}{B} \sum_{\boldsymbol{t} \sim \mathcal{D}} \sum_{j=1}^{N} \nabla_\phi [0 \log \rho_j + 1 \log(1 - \rho_j)] = \phi + \frac{\alpha}{B} \sum_{\boldsymbol{t} \sim \mathcal{D}} \sum_{j=1}^{N} \nabla_\phi \log(1 - \rho_j)
\tag{7}
$$

where

$$
\rho_j \overset{D}{=} P(Z(\boldsymbol{x}, \pi_\phi(\boldsymbol{x})|\Theta_{\boldsymbol{\psi_1}}) \in \{z_1, z_2, \cdots, z_j\})
\tag{8}
$$

### 4.4 EXPLORATION

Policy improvement constitutes a fundamental component of actor learning; similarly, exploitation also plays an important role. This study introduces a heuristic exploration method inspired by failure, which harnesses nociceptive information to enhance policy exploration strategies.

**Definition 4.1.** Given an action distribution $\hat{A} = \pi(\boldsymbol{x})$, the action entropy is defined as:

$$
\mathcal{H}(\hat{A}) \overset{D}{=} - \sum_{i=1}^{n} \sum_{j=1}^{m} p_{ij}(\boldsymbol{x}) \log p_{ij}(\boldsymbol{x})
\tag{9}
$$

Additionally, $\mathcal{H}(\hat{A})$ has a calculable upper bound:

$$
\overline{\mathcal{H}} \overset{D}{=} n \log m \geq \mathcal{H}(\hat{A}) \quad \forall \pi : \mathcal{X} \to \mathbb{R}^{n \times m}
\tag{10}
$$

Our objective is to correlate the action entropy with confidence levels. Specifically, increase the action entropy $\mathcal{H}(\hat{A})$ when there is a higher probability occurrence at lower discretization atoms within the discrete value distribution. To achieve this, we introduce an entropy exploration term. The proposed update rule for the actor is as follows:

$$
\phi \leftarrow \phi + \frac{\alpha\beta}{B} \sum_{\boldsymbol{t} \sim \mathcal{D}} s \nabla_\phi \frac{\mathcal{H}(\pi_\phi(\boldsymbol{x}))}{\overline{\mathcal{H}}}
$$

$$
s = \begin{cases} 1 & if \quad \max_{1 \leq j \leq N} \frac{N-j}{N-1} h \rho_j \geq \frac{\mathcal{H}(\pi_\phi(\boldsymbol{x}))}{\overline{\mathcal{H}}} \\ 0 & \text{otherwise} \end{cases}
\tag{11}
$$

where $\rho_j$ is same to in Equ equation 8 , $\beta > 0$ is the coefficient for the entropy term, $0 < h \leq 1$ regulates the scale of action entropy. An action entropy threshold, $\frac{N-j}{N-1} h \rho_j$, is assigned to each discrete atom of the value distribution such that the entropy exploration term will only activate when the action entropy $\mathcal{H}(\pi_\phi(\boldsymbol{x}))$ falls below this threshold. This threshold decreases as $j$ increases, which means that atoms of higher values have lower thresholds.

We also use the cumulative distribution $\rho_j$ to represent the confidence level of the agent with respect to the current state $\boldsymbol{x}$. It should be noted that for the $j^{\text{th}}$ value atom of a high-confidence agent, $\rho_j$ should be a small scalar since it represents the probability between the $1^{\text{th}}$ atom and the $j^{\text{th}}$ atom, which

is the lower value range. We use $\rho_j$ to correct the action entropy $\mathcal{H}(\pi_\phi(\boldsymbol{x}))$, so the low-confidence agent will increase it to seek various solutions with respect to state $\boldsymbol{x}$, however, the high-confidence one will not.

Integrating this with the prior section's material, the comprehensive update rule for the actor is:

$$\boldsymbol{\phi} \leftarrow \boldsymbol{\phi} + \frac{\alpha}{B} \sum_{\boldsymbol{t} \sim \mathcal{D}} \sum_{j=1}^{N} \nabla_\phi \log(1 - \rho_j) + \frac{\alpha\beta}{B} \sum_{\boldsymbol{t} \sim \mathcal{D}} s \nabla_\phi \frac{\mathcal{H}(\pi_\phi(\boldsymbol{x}))}{\overline{\mathcal{H}}} \tag{12}$$

During data collection in the training phase, the model also keeps track of the most recent time when each action atom is executed. In half of the episodes, before actions are executed, for each of the action dimensions, with a given small probability, the model replaces the action atom with the atom that is not executed for the longest time. The typical probability we chose is about $0.05/n$ where $n$ is the number of action dimensions.

## 5 EXPERIMENTS

We tested our model on continuous control of several tasks, including BipedalWalkerHardcore-v3 and MuJoCo tasks, and evaluated the performance of this discrete model. We also took SAC, TD3, and TQC, which are popular off-policy algorithms for continuous tasks, as baselines and applied them to the same tasks for comparisons. Experiments are conducted on a desktop workstation with the Intel® Core ™i9-12900 Processor, 64GB RAM, and the NVIDIA® GeForce RTX ™4090.

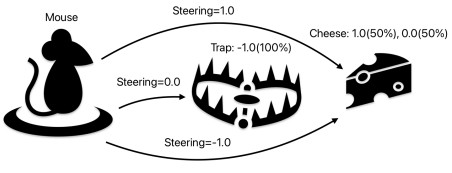
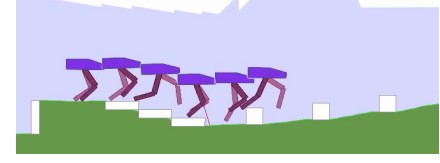

| (a) A trap or cheese problem. | (b) The BipedalWalkerHardcore-v3 task. |

Figure 5: Two of the tasks. (a) Averaging two good choices results in a bad choice. Our model does not make this mistake. (b) The robot parkours with our model.

### 5.1 TRAP OR CHEESE PROBLEM

We designed a toy task called "Trap or Cheese" to illustrate that continuous models averaging good actions can result in a bad action, but our model does not have this problem. As shown in Figure 5(a), there is a trap in front of the mouse, behind which lies a piece of cheese. When the mouse chooses to move straight ahead, it falls into the trap and dies, resulting in a reward of -1.0. When the mouse chooses to turn left or right, it can bypass the trap and reach the cheese. However, there is a 50% chance that the cheese has expired and cannot be eaten, resulting in a reward of 0.0. If the cheese is not expired, the reward is 1.0. Obviously, a normal mouse would not choose to walk into the trap.

Both SAC and our model are tested in this task. The results show that SAC tends to unhesitatingly choose the middle route and walk into the trap, with an average score staying at -1.0. In contrast, our discrete can learn the correct strategy, with an average score staying at 0.5. This simple task is difficult for SAC because, although its critic network can learn that moving forward is a very bad choice, since moving forward can be considered as an average of moving left and right, SAC still chooses to move forward. This problem could widely exist in continuous RL models which tend to average best actions. The BipedalWalkerHardcore task shares a similar property when stepping over obstacles. Hence, we suspect it is the reason why continuous model cannot solve this task as well as our model. We will discuss it further in later sections.

### 5.2 BIPEDALWALKERHARDCORE-V3

The BipedalWalkerHardcore-v3 (Towers et al., 2023) task is one of the benchmarks in OpenAI Gym. Compared to other motion control tasks in OpenAI Gym, the challenges in BipedalWalkerHardcore-

v3 are from its variation of terrains, partial observability, and high penalty when falling. The task's objective is to control the joints of a planar bipedal robot to walk through complex terrains involving randomly generated obstacles like ladders, stumps, and pitfalls. An agent has to find and learn different motions for each type of obstacle. The robot's observation of the environment is by a LIDAR that only returns 10 Lidar rangefinder measurements of the immediate terrain, hence the environment is partially observable by the agent. The task also provides substantial penalties for robot falls simulating "nociceptive stimuli", which disturbs convergence of the critic network, and potentially hinders skill acquisition. As per (Wei and Ying, 2021), adapting TD3(Fujimoto et al., 2018) to BipedalWalkerHardcore-v3 necessitates reducing these penalties.

We trained our model and baselines on the BipedalWalkerHardcore-v3 task for 20 million time steps. Figure 6 left shows the reward returns during training. Our model outperformed the baseline models. In an evaluation configuration, our model achieved a mean score of 324.8 in 10,000 trials. We present the training curves of our model and baseline models in Figure **??** and provide a summary of their final evaluation results in 10,000 trials in Table 1.

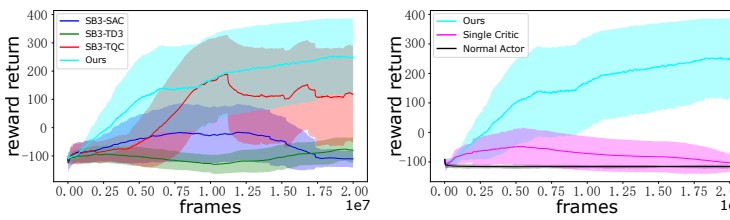

Figure 6: The rewards during training in BipedalWalkerHardcore-v3 task. Left: Comparison between our model and baselines. Right: Ablation experiments.

Table 1: BipedalWalkerHardcore-v3 Evaluation

| Task | Ours | SB3-SAC | SB3-TD3 | SB3-TQC |
|---|---|---|---|---|
| BipedalWalkerHardcore-v3 | $324.8 \pm 31.6$ | $5.1 \pm 97.3$ | $-20.1 \pm 22.4$ | $217.9 \pm 120.3$ |

To understand the necessity for each module, we performed ablation studies on BipedalWalkerHardcore-v3 (Figure 6 right). We disabled our twin critic networks, the outcomes of which are depicted by the curve labeled "Single Critic". Following D3PG (Barth-Maron et al., 2018), we substituted our discrete actor with a conventional continuous action actor based on our twin critic network. The results are depicted by the curve labeled "Normal Actor". The results suggest that the different modules proposed in our model are necessary for the model's performance.

### 5.3 MuJoCo

Although during building our model, we mainly tested it on a task with strong reward outliers and changing environments that need risky actions, we would also evaluate it in typical continuous control tasks that are more consistent. Experiments were conducted within MuJoCo (Towers et al., 2023) on a series of tasks, including Ant, HalfCheetah, Hopper, Humanoid, and Walker2D. These tasks are to control a corresponding type of robot for a forward motion. Because the environment is just level terrain, observation of the environment around the robot is not necessary. Hence, the state of the task is fully accessible with proprioceptors, odometry, and accelerometers as the task provided, and the tasks exhibit a pronounced Markovian property.

Although discretizing a continuous action space can decrease control precision and increase the difficulty of problem-solving, we did not observe a cliff-like drop in the performance of the model proposed in this paper on MuJoCo tasks. However, its learning efficiency may decrease, requiring more data and training steps. In Figure 7, we present the training curves of the proposed model in the MuJoCo tasks. The light-shaded area represents the scores obtained by the model during training and exploration, while the solid line depicts the average scores obtained by the model over 100 tests in the testing environment during training. Due to the random selection of actions and the addition of noise

Table 2: MuJoCo Evaluation. The data of SAC, TD3, TQC is from (Kuznetsov et al., 2020).

| Task | Ours | SAC | TD3 | TQC |
|------|------|-----|-----|-----|
| Ant | 6988 | 6160 | 5680 | 8010 |
| HalfCheetah | 13800 | 12410 | 15120 | 18090 |
| Hopper | 4118 | 2860 | 3310 | 3710 |
| Humanoid | 9992 | 7760 | 5400 | 9540 |
| Walker2D | 5775 | 5760 | 5110 | 7030 |

for exploration during training, the variance in scores is relatively large. However, during testing on the evaluation set, we always choose the action with the highest probability output by the actor. Therefore, the scores obtained during testing are often better than those obtained during training.

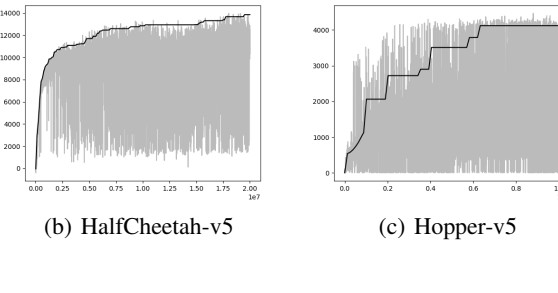

(a) Ant-v5     (b) HalfCheetah-v5     (c) Hopper-v5

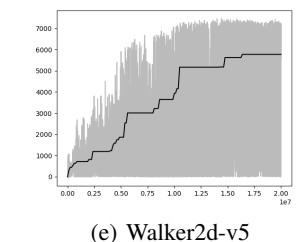

(d) Humanoid-v5     (e) Walker2d-v5

Figure 7: MuJoCo Training return

## 6 CONCLUSION & DISCUSSION

In this paper, we proposed an off-policy Deep RL model with a stochastic discrete actor and critics. By approximating continuous values and actions with discrete distributions, this model can capture multimodal distribution in action and value spaces and solves BipedalWalkerHardcore-v3 with the state-of-the-art performance (see the (OpenAI) Leaderboard). The model also has performance near to baselines such as TQC, TD3, and SAC in various tasks.

Through this model and experiments, our curiosity about whether a discrete RL model can learn continuous tasks is confirmed. Given discrete control signals can be applied to real-world control systems, a RL model with discrete action space can also be applied to continuous control tasks. Although action resolution become lower, with the new ability to explore and learn multimodal action distribution, as well as distributional-RL-style critics, the model can solve a task with strong punishments and HRHR actions.

We noticed that, in BipedalWalkerHardcore-v3 task, the reward returns of TD3 and SAC are much lower than TQC and our model. A possible reason is that TD3 and SAC failed to predict return by high-risk-high-return (HRHR) actions, such as climbing over a box. A successful HRHR action will result in a high return, but a failed one will result in a high punishment, which is much worse than doing nothing. Hence the returns follow a multimodal distribution. TD3 and SAC focus on modeling the scalar expectation but not the distribution of returns. Hence, it is can similar situation to the

"Trap or Cheese" problem. Models using scalar return expectation tend to believe if a high reward is conditioned on two different actions, it is also conditioned on the middle of actions. Our model uses discrete return expectations and does not have this problem.

Our model also benefits from the concept of discrete actions. In BipedalWalkerHardcore-v3, it successfully learned a realistic and human-like gait without modification of the reward functions, whereas baselines tend to master a gait similar to quadrupeds. One reason is that our model explores the action space with a discrete distribution, which can approximate a multimodal distribution. For example, when the walker tries to step over a large stump, it is better to either jump or maintain balance conservatively. A moderate walk likely results in a meaningless fall. Typical models with continuous actors, such as TQC, adopt Gaussian distribution or the Ornstein-Uhlenbeck process, are limited in action exploratory and its refinement, and tend to master a limping gait.

The stochastic discrete actor also allows for expansive and efficacious exploration of the action space and facilitates the acquisition of a natural, harmonious gait resembling human bipedal alternating stepping. Furthermore, the model displays adaptive gait transitions when faced with a diverse array of challenging obstacles, mitigating the risk of toppling; for instance, opting for energy-efficient alternating stepping on even terrain and ascending ladders, transitioning to a safer gait when descending ladders, modifying its step length for minor stumps, and initiating climbing maneuvers for large stumps. Allowing multiple gaits in the same situation helps in exploration and learning in complex environments, so a robot can switch between different motion primitives, or gaits, instead of forgetting one to relearn another one.

As a preliminary trial to apply discrete actions to RL models for continuous tasks, our model may not achieve the best performance. A possible improvement of this work is interpolation between discrete actions. That is, using a kernel to map multiple discrete actions to a continuous action, could improve the resolution of the actions. As shown in experiments and discussed, our model is suitable for robot tasks with complex interactions with the environment. In real-world tasks, environments are usually complex and robots need to interact with them. Hence, this model can potentially facilitate the application of robots in the real world. It is worth investigating in the future.

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

# A    MATH SYMBOLS

Table 3: Mathematical Symbols

| Symbol | Description | Typical value |
|---|---|---|
| $\gamma$ | Discount factor. | 0.98 or 0.99 |
| $V_{MAX}$ | Upper bound of the discrete value. | $\frac{1}{1-\gamma}$ |
| $V_{MIN}$ | Lower bound of discrete value. | $-\frac{1}{1-\gamma}$ |
| $Z$ | Random variable for discrete value. | |
| $z_i$ | The $i^{\text{th}}$ discrete value atom of $Z$. | $[V_{MIN}, V_{MAX}]$ |
| $\boldsymbol{x}$ | Sample of current state observation. | |
| $\boldsymbol{a}$ | An action sample vector. | |
| $\hat{\boldsymbol{a}}$ | Distribution of an action vector. | |
| $\hat{A}$ | An action distribution matrix of a multidimensional discrete action space, in which the sum of each row is 1. | |
| $A$ | An action sample matrix. Each row in $A$ adopts one-hot coding sampled from the corresponding row in $\hat{A}$. | |
| $\hat{A}'$ | Action distribution for the next state. | |
| $r$ | Reward from an environment. | $\leq 1$ |
| $\boldsymbol{x}'$ | Sample of next state observation. | |
| $\hat{\boldsymbol{x}}'$ | Distribution of the next state observation. | |
| $\hat{\cdot}$ | Distribution of a random variable. | |
| $\cdot'$ | A variable in the next time step. | |
| $\stackrel{D}{=}$ | Denotes definition. | |
| $\mathcal{X}$ | Space of state observations. | |
| $\mathcal{A}$ | Space of actions. | |
| $\left\lceil \frac{1}{1-\gamma} \right\rceil$ | Ceiling of $\frac{1}{1-\gamma}$. | |
| $N$ | Number of discrete atoms for $Z$. | 51 |
| $\hat{\mathcal{T}}Z$ | $r + \gamma Z'$. | |
| $\Phi\hat{\mathcal{T}}Z$ | Projecting $\hat{\mathcal{T}}Z$ back to origin discrete value atoms. | |
| $\tilde{Z}$ | The estimated Z from the twin critic networks. | |
| $\Theta$ | Discrete distribution critic network. | |
| $(\cdot)_i$ | $i^{\text{th}}$ Element of a Vector. | |
| $\pi$ | Policy for action selection. | |
| $Q$ | Expected scalar critic network. | |
| $\boldsymbol{\psi_1}, \boldsymbol{\psi_2}$ | Parameters of first and second critic networks. | |
| $\boldsymbol{\phi}$ | Parameters of actor network | |
| $\boldsymbol{\psi_1'}, \boldsymbol{\psi_2'}, \boldsymbol{\phi}'$ | Parameters for delayed updated networks. | |
| $\leftarrow$ | Denotes parameter update. | |
| $\nabla_{\boldsymbol{\omega}} J$ | Gradient of $J$ with respect to $\boldsymbol{\omega}$. | |
| $\alpha$ | Learning rate. | $\leq 10^{-3}$ |
| $B$ | Batch size. | 256 or 512 |
| $\boldsymbol{t} \sim \mathcal{D}$ | Sample from Replay Buffer. | |
| $n$ | Number of dimensions in action. | $\leq 20$ |
| $m$ | Number of atoms per action dimension. | 51 |
| $\mathcal{H}(\hat{A})$ | Entropy of action $\hat{A}$. | $\leq n \log m$ |
| $\overline{\mathcal{H}}$ | Maximum entropy of action. | $n \log m$ |
| $h$ | Scaling factor for action entropy. | 0.5 |
| $\beta$ | Coefficient for exploration. | 0.5 |
| sup | Represents the upper bound. | |

# B    MATHEMATICAL ANALYSIS OF TRAP CHEESE PROBLEM

We describe the $Q$ function of the Trap Cheese problem as:

$$Q(x_0, a) = \begin{cases} 0.5 & if \quad a \in [-1 - \delta, -1 + \delta] \cup [1 - \delta, 1 + \delta] \\ -1 & otherwise \end{cases} \tag{13}$$

Where $\delta$ is used to denote the width of the range where high rewards can be obtained, $0 < \delta < 1$. For convenience, we represent this region with the symbol $\mathcal{C}(\delta)$. We are interested in the maximum likelihood estimation of the normal distribution $a \sim N(\mu, \sigma^2)$ on $\mathcal{C}(\delta)$.

$$
\begin{aligned}
\log L &= \int_{\mathcal{C}(\delta)} \log\left(\frac{1}{\sqrt{2\pi}\sigma} e^{-\frac{(a-\mu)^2}{2\sigma^2}}\right) da \\
&= -4\delta \log(\sqrt{2\pi}\sigma) - \frac{1}{2\sigma^2} \int_{\mathcal{C}(\delta)} (a-\mu)^2 da \\
&= -4\delta \log(\sqrt{2\pi}\sigma) - \frac{1}{6\sigma^2}[(1+\delta-\mu)^3 - (1-\delta-\mu)^3 + (-1+\delta-\mu)^3 - (-1-\delta-\mu)^3] \\
&= -4\delta \log(\sqrt{2\pi}\sigma) - \frac{1}{6\sigma^2}[6(1-\mu)^2\delta + 2\delta^3 + 6(1+\mu)^2\delta + 2\delta^3] \\
&= -4\delta \log(\sqrt{2\pi}\sigma) - \frac{2\delta}{3\sigma^2}[3(1+\mu^2) + \delta^2]
\end{aligned}
\tag{14}
$$

Letting $\frac{\partial \log L}{\partial \mu} = 0$ and $\frac{\partial \log L}{\partial \sigma} = 0$, we can obtain the maximum likelihood estimates for $\mu$ and $\sigma$.

$$
\begin{aligned}
\frac{\partial \log L}{\partial \mu} &= -\frac{4\delta\mu}{\sigma^2}, \quad \tilde{\mu} = 0 \\
\frac{\partial \log L}{\partial \sigma} &= -\frac{4\delta}{\sigma} + \frac{4\delta}{3\sigma^3}[3(1+\mu^2) + \delta^2], \quad \tilde{\sigma}^2 = 1 + \frac{\delta^2}{3}
\end{aligned}
\tag{15}
$$

Let's verify whether $\tilde{\mu} = 0$ and $\tilde{\sigma}^2 = 1 + \frac{\delta^2}{3}$ is the unique critical point by computing its Hessian matrix first.

$$
\begin{bmatrix} \frac{\partial^2 \log L}{\partial \mu^2} & \frac{\partial^2 \log L}{\partial \mu \partial \sigma} \\ \frac{\partial^2 \log L}{\partial \mu \partial \sigma} & \frac{\partial^2 \log L}{\partial \sigma^2} \end{bmatrix} = \begin{bmatrix} -\frac{4\delta}{\sigma^2} & \frac{8\delta\mu}{\sigma^3} \\ \frac{8\delta\mu}{\sigma^3} & \frac{4\delta}{\sigma^4}(\sigma^2 - \delta^2 - 3) \end{bmatrix}
\tag{16}
$$

Substituting $\tilde{\mu} = 0$ and $\tilde{\sigma}^2 = 1 + \frac{\delta^2}{3}$, we obtain:

$$
\begin{bmatrix} \frac{\partial^2 \log L}{\partial \mu^2} & \frac{\partial^2 \log L}{\partial \mu \partial \sigma} \\ \frac{\partial^2 \log L}{\partial \mu \partial \sigma} & \frac{\partial^2 \log L}{\partial \sigma^2} \end{bmatrix}_{\tilde{\mu}, \tilde{\sigma}} = \begin{bmatrix} -\frac{4\delta}{1+\frac{\delta^2}{3}} & 0 \\ 0 & -\frac{8\delta}{1+\frac{\delta^2}{3}} \end{bmatrix} \preceq 0
\tag{17}
$$

Therefore, $\tilde{\mu} = 0$ and $\tilde{\sigma}^2 = 1 + \frac{\delta^2}{3}$ is the unique maximum point on the domain. Although $N(\tilde{\mu}, \tilde{\sigma}^2)$ is the maximum likelihood estimate for the set $\mathcal{C}(\delta)$ under the assumption of a normal distribution, its maximum probability density point $\tilde{\mu}$ does not yield satisfactory values on the $Q$ function; evidently, $Q(x_0, \tilde{\mu}) = -1$. Now we will compute the maximum likelihood estimate again, this time on a discrete distribution.

$$
P(a = a_i) = p_i, \quad i = 1, 2, \cdots, m, \quad \sum_i^m p_i = 1.0, \quad p_i >= 0
\tag{18}
$$

In the case of a discrete distribution, the range of action $a$ is given by $\mathcal{A} = \{a_1, a_2, \cdots, a_m\}$, and $\mathcal{A} \cap \mathcal{C}(\delta) \neq \emptyset$.

$$
L = \prod_{\mathcal{A} \cap \mathcal{C}(\delta)} pi
\tag{19}
$$

According to AM–GM inequality, we have:

$$\prod_{\mathcal{A} \cap \mathcal{C}(\delta)}^{\|\mathcal{A} \cap \mathcal{C}(\delta)\|} \sqrt{\prod_{\mathcal{A} \cap \mathcal{C}(\delta)} pi} \leq \frac{\sum_{\mathcal{A} \cap \mathcal{C}} p_i}{\|\mathcal{A} \cap \mathcal{C}(\delta)\|} \leq \frac{1}{\|\mathcal{A} \cap \mathcal{C}(\delta)\|} \tag{20}$$

The two equalities in the above inequality can be attained; therefore, the maximum likelihood estimate in the case of a discrete distribution is:

$$\tilde{p}_i = \begin{cases} \frac{1}{\|\mathcal{A} \cap \mathcal{C}(\delta)\|} & if \quad a_i \in \mathcal{C}(\delta) \\ 0 & otherwise \end{cases} \tag{21}$$

In the maximum likelihood estimate of a discrete distribution, we take the point $a_k$ with the highest probability, and obviously it satisfies $Q(x_0, a_k) = 0.5$. Based on the above discussion, we can infer that when dealing with complex obstacles, discrete distributions might have an advantage over normal distributions, at least in the context of maximum likelihood estimation.

## C  IMPLEMENTATION DETAILS

### C.1  REWARD NORMALIZATION

Reward Normalization is crucial in the training and convergence of models. The original reward function, denoted as $R_1(\boldsymbol{x}, A)$, is advised to be transformed into a normalized form $R_2(\boldsymbol{x}, A)$, which ideally possesses the following characteristics:

$$R_2(\boldsymbol{x}, A) = C R_1(\boldsymbol{x}, A), \quad C > 0, \quad \sup_{\boldsymbol{x}, A} R_2(\boldsymbol{x}, A) \leq 1 \tag{22}$$

If a constant $C$, typically represented as $\frac{1}{\sup_{\boldsymbol{x}, A} R_1(\boldsymbol{x}, A)}$, can be identified, the following equation holds:

$$\begin{aligned} Z_t &= R_2(\boldsymbol{x_t}, A_t) + \gamma R_2(\boldsymbol{x_{t-1}}, A_{t-1}) + \gamma^2 R_2(\boldsymbol{x_{t-2}}, A_{t-2}) + \cdots \\ &\leq 1 + \gamma 1 + \gamma^2 1 + \cdots \leq \frac{1}{1-\gamma} \end{aligned} \tag{23}$$

Taking into account the upper bound mentioned above, we recommend configuring the hyperparameters $V_{MAX} = \frac{1}{1-\gamma}$.

### C.2  LOGARITHMIC OPERATIONS

If logarithmic operations are directly used to compute the loss function, it will result in significant precision loss, especially when dealing with very small values. Therefore, directly using the logarithm operator is unwise; we need to make some transformations on paper to avoid these precision losses. The technique demonstrated below is the "log sum exp" trick.

$$\log\left(\sum_{1 \leq i \leq N} e^{x_i}\right) = x^* + \log\left(\sum_{1 \leq i \leq N} e^{x_i - x^*}\right), \quad x^* = \max_{1 \leq i \leq N} x_i \tag{24}$$

The above transformation ensures that the values inside the logarithmic operations are greater than 1, thereby avoiding the problem of significant precision loss when the values are very small. Based on the above discussion, "log softmax" can be represented as:

$$\log\left(\frac{e^{x_j}}{\sum_{1 \leq i \leq N} e^{x_i}}\right) = x_j - x^* - \log\left(\sum_{1 \leq i \leq N} e^{x_i - x^*}\right), \quad x^* = \max_{1 \leq i \leq N} x_i \tag{25}$$

Furthermore, for the logarithmic operation of cumulative distribution, it can be represented as:

$$\log(1 - \frac{\sum_{1 \leq i \leq K} e^{x_i}}{\sum_{1 \leq i \leq N} e^{x_i}}) = \log(\frac{\sum_{K < i \leq N} e^{x_i}}{\sum_{1 \leq i \leq N} e^{x_i}})$$

$$= (x^{**} + \log(\sum_{K < i \leq N} e^{x_i - x^{**}})) - (x^* + \log(\sum_{1 \leq i \leq N} e^{x_i - x^*})) \quad (26)$$

$$x^* = \max_{1 \leq i \leq N} x_i, \quad x^{**} = \max_{K < i \leq N} x_i$$

In practice, we found that setting a near-zero lower bound (such as $\epsilon = 0.0001$) for all cumulative probabilities when constructing the Policy Loss will be more robust. This helps prevent the actor network from making significant policy changes in pursuit of minor fluctuations in noise.

$$\log(1 - (1 - \epsilon)\frac{\sum_{1 \leq i \leq K} e^{x_i}}{\sum_{1 \leq i \leq N} e^{x_i}})$$

$$= \log(\frac{\epsilon \sum_{1 \leq i \leq K} e^{x_i} + \sum_{K < i \leq N} e^{x_i}}{\sum_{1 \leq i \leq N} e^{x_i}})$$

$$= \log(\frac{\sum_{1 \leq i \leq K} e^{x_i + \log(\epsilon)} + \sum_{K < i \leq N} e^{x_i}}{\sum_{1 \leq i \leq N} e^{x_i}}) \quad (27)$$

$$= x^{**} + \log(\sum_{1 \leq i \leq K} e^{x_i + \log(\epsilon) - x^{**}} + \sum_{K < i \leq N} e^{x_i - x^{**}}) - (x^* + \log(\sum_{1 \leq i \leq N} e^{x_i - x^*}))$$

$$x^* = \max_{1 \leq i \leq N} x_i, \quad x^{**} = \max(\max_{1 \leq i \leq K} x_i + \log(\epsilon), \max_{K < i \leq N} x_i)$$

