# OpenReview forum: "A Discrete Actor and Critic for Reinforcement Learning on Continuous Tasks"
_ICLR.cc/2025/Conference — Submitted to ICLR 2025_

### Official Review · Reviewer_PyGc · 2024-10-27

**Soundness:** 1
**Presentation:** 2
**Contribution:** 1
**Rating:** 3
**Confidence:** 4

**Summary:**

The paper proposes a discrete actor-critic method that discretizes continuous action spaces via a decoupled actor and leverages distributional double Q-learning for stable training. The authors motivate the approach via pulse-width-modulation and apply their approach with a fine-grained action distribution of 51 bins per dimension on several benchmark tasks against a selection of continuous control baselines. The approach performs favorably, indicating that discretized control may be a promising alternative for domains traditionally solved with continuous policies, e.g., robotics.

**Strengths:**

- The paper investigates an interesting problem with real-world implications for exploration and learning in continuous robot control tasks.
- The procedure for clipped distributional double Q-learning and it's impact on performance in Figure 6 is an interesting insight.
- Comparison to strong baseline agents on a variety of tasks is helpful to put the proposed method into perspective

**Weaknesses:**

- The paper investigates the interesting topic of discretizing continuous control tasks to facilitate efficient learning control, but overlooks a line of key related work. Decoupled distributional critics for discretized control were shown to be competitive with continuous actor-critic approaches in [1], with prior motivation for PWM-type bang-bang control in [2] and decoupled Q-learning approaches for single agent control in [3-4], with additional application to fine-grained discretization as well as hardware control [5-6].
- The method presented here uses multiple concept also presented/employed in references [1-6] (and references therein). Without a rigorous discussion, it is difficult to judge which parts of the presented method may qualify for novel contribution.
- Figure 1 appears to be a slight modification without attribution of the PWM figure from Wikipedia (https://en.wikipedia.org/wiki/Pulse-width_modulation#/media/File:PWM,_3-level.svg) - could you clarify how this figure was created?
- Table 3 indicates a 51-bin discretization per action dimension. The transition from the PWM motivation (=2/3 bins) to this fine-grained discretization could be smoother.
- Additional proofreading would be beneficial (e.g., line 316: “… is same to in Equ equation 8 ,” ; line 390: “Figure ??”)
- The “Trap or Cheese” experiment would profit from additional quantitative as well as qualitative analysis. It is not immediately clear why SAC would average options and not commit to one of the modes in an RL setting in practice (e.g., always select left / always select right).
- The SAC / TD3 / TQC baselines from Table 2 should be added to Figure 7. Are results in Table 2 compared at the same #frames for each algorithm? For example, TQC runs the Ant - Walker tasks for 5/5/3/10/5 x 1e6 max frames in their Figure 5, while Figure 7 here runs experiments for more than 1e7 frames.
- Kuznetsov et al. use OpenAI Gym-v3 environments, vs the experiments in Figure 7 use Gym-v5. It should be justified why these versions are 1-to-1 comparable (if they are).

**References:**

[1] T. Seyde, P. Werner, W. Schwarting, I. Gilitschenski, M. Riedmiller, D. Rus, and M. Wulfmeier. "Solving continuous control via q-learning." ICLR, 2023.

[2] T. Seyde, I. Gilitschenski, W. Schwarting, B. Stellato, M. Riedmiller, M. Wulfmeier, and D. Rus. "Is bang-bang control all you need? solving continuous control with bernoulli policies." NeurIPS, 2021.

[3] A. Tavakoli, F. Pardo, and P. Kormushev. "Action branching architectures for deep reinforcement learning." AAAI, 2018.

[4] A. Tavakoli, M. Fatemi, and P. Kormushev. "Learning to represent action values as a hypergraph on the action vertices." ICLR, 2021.

[5] D. Ireland, and G. Montana. "Revalued: Regularised ensemble value-decomposition for factorisable markov decision processes." ICLR, 2024.

[6] Y. Seo, J. Uruç, and S. James. "Continuous control with coarse-to-fine reinforcement learning." CoRL, 2024.

**Questions:**

- See also weaknesses above for implicit questions
- How many seeds were the experiments in Figures 6 and 7 averaged over?
- Is the impact of removing the double critic not a bit concerning w/r/t training stability? A comparison to the impact of single vs double critic on other baselines would be insightful.
- What is the motivation for randomly selecting actions that have not been selected for the longest time during half of the episodes? Did this improve performance over only using the entropy exploration method?

---

### Official Review · Reviewer_brmt · 2024-10-30

**Soundness:** 2
**Presentation:** 2
**Contribution:** 2
**Rating:** 3
**Confidence:** 3

**Summary:**

This work presents a discrete-actor for continuous control tasks.
They show competitive performance on a range of standard benchmark environments.

**Strengths:**

Overall the method looks promising.
There is a good motivation for using discrete representations, especially around improved exploration in large continuous spaces.

**Weaknesses:**

Experiments focus on simple environments. The major challenges occur with large action spaces and where the differences in run-time, sample efficiency, etc... become more noticible.
For example, humanoids such as in Adversarial Motion Priors or Perpetual Humanoid Controller, have up to 70 degrees of freedom.

**Questions:**

1. How would this method compare with very large action spaces?

2. It seems there's no structure in the action space. One strong benefit of continuous representations is that there is a notion of close-ness and order. Could this be applied here? Why not do so?

---

### Official Review · Reviewer_rmKX · 2024-11-02

**Soundness:** 2
**Presentation:** 2
**Contribution:** 1
**Rating:** 1
**Confidence:** 4

**Summary:**

This paper presents a technique to deal with discredited continuous high-dimensional action spaces.
To overcome the curse of dimensionality, the authors choose a factorized policy representation, where each action dimension is discretized, and use distributional RL to learn the critic and improve the policy.

Discretized policies can capture multimodal distributions (in contrast to more classic Gaussian policies), providing a more effective exploration.

**Strengths:**

Whether actions should be represented with discrete or continuous variables is very compelling. Each of the two representation have their own advantages and disadvantages. Discrete actions, in contrast with continuous ones, for example, naturally allow for multimodal exploration. On the other hand, continuous actions more easily allow for generalization, and they do not suffer particularly the curse of dimensionality (which is why they have been so largely employed in RL).

Originality
--------------

The algorithm proposed is novel.

Quality and Clarity
-------------------------

The method is described clearly.

Significance
---------------

The paper has the potential to address a very interesting question about whether (or when) discretizing actions is convenient.

**Weaknesses:**

Main Weaknesses
------------------------

This paper lacks a central research question: this is a real issue because I cannot understand what the authors are trying to achieve with their algorithm. I think this could be solved by clearly stating what is the problem that the authors want to solve (or mitigate) or what is the main research question.

From the abstract, it seems that the authors want to develop a novel action discretization mechanism, and they mention that they draw inspiration from Pulse-With-Modulation (PWM) from classic control. But in the method, nothing resembles PWM (unless I've missed the connection).

Discretization does not play a central role in the paper either: the discretization is provided as hyperparameters, and what the authors propose is how to represent the new discrete action space (in particular, with a rather classic soft-max parametrization).

So, I am left unsure of what the real contribution of the paper is.

In the paper's conclusion, the authors highlight that their method provides a multimodal exploration, but my question is: do we need to discretize the action space to achieve it? Normalizing flows, diffusion processes, and mixtures of Gaussian (as the output of the policy network) achieve that as well... Discrete distributions are, by definition, multimodal (one mode for discrete action)... what is special in it?

I am also confused by the algorithm proposed by the authors: it is unclear why they chose distributional RL versus classic RL... I have nothing against distributional RL (quite the opposite), but I don't understand if this choice has a particular reason. It is the same about double Q-learning and clipping values (which are techniques to prevent the overestimation of the Q-function)... What is their role in this paper? The same goes for the entropic bonus defined in Eq. 11.

In summary, the proposed method seems like a "collection" of methods put together but without a clear objective. Perhaps the paper will become clearer when the authors better define their goal or research question.

Experiments
-----------------

**Trap-Or-Cheese Problem**: "Models averaging good actions can result in a bad action."... I would say this is a true statement, but that is not due to continuous policies but rather to how the distribution is defined. Gaussian policies struggle in the presented situation because they have only one mode. That is not to say we need to discretize the action space to make the policy multimodal; other methods can be used. Furthermore, comparing SAC with the proposed algorithm is not meaningful, as the proposed algorithm uses distributional RL while SAC does not. (P.S., DQN, PPO, etc, can handle discrete action spaces too, thus, multimodality). **I can't see in the paper the results of this experiment**

**Bipedal Walker** How many discretized actions were used?

**Mujoco** What are the confidence intervals of Table 2? What do I see in figure 7?? What is the difference between gray and black lines?

**In general, the algorithms' hyperparameters (neural network site, discretization, learning rates..) are completely missing.*

Justification for the grades
-----------------------------------

**Soundess: Fair**. It is really hard to judge if the paper is sound because the objective of the paper is unclear.
**Presentation: Fair **. Even though the paper is generally well written in terms of English/grammar, and the method is explained somewhat clearly, the presentation is not generally good because of the lack of a main research question, making it difficult for the reviewer to judge the method, and to make a "take-home message".... What did I learn reading this paper?
**Contribution: Poor.** As of now, I do not see a novel contribution in this paper: action discretization has already been explored in the community with better methods (in this paper, the action discretization comes as a hyperparameter); I do not see any novelty relevant in the use of distributional reinforcement learning and in the modified entropic objective (entropic bonuses are quite used in standard RL, see SAC).


Minor Typos / Unclear Points (Not relevant for the scores of this review)
----------------------------------------------------------------------------------------------

* Line 316, "Equ equation"
* Line 390, "Figure ??"
* Figure 6: are shaded areas in STD or confidence intervals?
* Line 198: It does not make sense to say that the elements of a set are independent. Dependency has a statistical meaning. For example, one can define a policy where the different dimensions of an action are sampled independently (as it would happen with a Gaussian with diagonal covariance), or they could be dependent (as it would happen with a Gaussian with non-diagonal covariance).

**Questions:**

1. What is the main research question of the paper? What do you try to achieve?
2. What do you do to achieve your research goal? How do the outlined components of your algorithm contribute to solving the research question?
3. Why do you compare with SAC, TD3, and TQC? They are all continuous actions. Shouldn't you compare with discrete RL as well? i.e., PPO with discrete policy or DQN? And what about algorithms that discretize the action space autonomously?
4. Can you provide more details on how SAC and your algorithm are compared in the Cheese-Trap environment (and the results)?
5. Can you clarify if the shaded areas in Figure 6 are STDs or confidence intervals?
6. What are the confidence intervals of Table 2?
7. Can you explain figure 7?
8. I am unsure whether *action discretization* or *multimodality* the focus of your work. And why *distributional RL* is relevant?
9. Please include a table of hyperparameters in the appendix.

---

### Official Review · Reviewer_TGuu · 2024-11-04

**Soundness:** 2
**Presentation:** 1
**Contribution:** 1
**Rating:** 3
**Confidence:** 4

**Summary:**

The paper proposes a reinforcement learning algorithm that discretize continuous action spaces and combines TD3 with C51. The authors demonstrate that their model can achieve state-of-the-art performance on the BipedalWalkerHardcore-v3 task and competitive results on various MuJoCo tasks.

**Strengths:**

- the trap and cheese problem is a nice illustration of the limitation of Gaussian policies

**Weaknesses:**

- The paper needs a major reviewing, it is full of typo
  - L36/L37 "However, by varying the ratio of the time discrete values are presented"
  - L48 "With idea, we proposed a model with discrete action"
  - "In contrast, our discrete can learn the correct strategy"

- Even though I understood the general idea of the paper, it is very hard to read and needs to be deciphered.
  - "For continuous tasks, such as motion control tasks, evaluating all possible action is not possible, hence RL models with discrete action space can suffer from the curse of dimensionality". The start of the sentence talks about continuous action space and the conclusion is about the curse of dimensionality of discrete action space methods...
  - "We investigated RL models with discrete action spaces with performance comparable to continuous models on continuous tasks"

- The paper cites relevant baselines but do not compare with them nor explain why. It should be compared to baselines that also go beyond simple Gaussian policies like [1].

- The combination of C51 with TD3 is quite an incremental contribution.


[1] Tang, Y., & Agrawal, S. (2020). Discretizing Continuous Action Space for On-Policy Optimization. Proceedings of the AAAI Conference on Artificial Intelligence, 34(04), 5981-5988. https://doi.org/10.1609/aaai.v34i04.6059

**Questions:**

1. How much multi-modal are the trained policies?

2. Are you sure that equation (7) is better than simply maximizing the expected Q value?

3. Why Fig. 7 contains only the results of the proposed method?

---

### Meta-Review · Area_Chair_oPVG · 2024-12-21

**Metareview:**

The paper aims at addressing continuous control through discretization of actions. The motivation is that in PWM-based control systems where discrete changes in voltage produce continuous control of current. Based on TD3, the authors use distributional RL to learn the critic.

While I understand the benefits of discretizing actions, for example, for maintaining multimodal distributions, the paper has to go a long way to clearly describe the benefits of this approach. One good way is to pose a research problem that can be motivated from first principles. For example, the authors could start by discussing the drawbacks of maintaining a continuous distribution and the advantages of maintaining discrete actions for learning policies. Positioning the paper in terms of a research problem helps both the authors and the readers to understand what the closest baseline algorithms would be. For example, is the inability of representing multimodal distributions through normal policies the main issue we are addressing in this paper? Then a class of other approaches becomes relevant such as Normalizing flows, diffusion processes, and mixtures of Gaussian, as mentioned by a reviewer. I strongly suggest rephrasing the work in terms of a solid research question for the next submission.

**Additional Comments On Reviewer Discussion:**

The reviewers also found the work to be undermotivated, lacking a central research question. Moreover, there are other issues as well such as chosen environments being too simple or not motivating different parts of the algorithm proposed. The authors did not respond during the rebuttal phase.

---

### Decision · Program_Chairs · 2025-01-22

Reject